# A Review of the Anti-Obesity Effects of Wild Edible Plants in the Mediterranean Diet and Their Active Compounds: From Traditional Uses to Action Mechanisms and Therapeutic Targets

**DOI:** 10.3390/ijms241612641

**Published:** 2023-08-10

**Authors:** Bashar Saad

**Affiliations:** 1Qasemi Research Center, Al-Qasemi Academic College, P.O. Box 124, Baqa al-Gharbiyye 3010000, Israel; bashar@qsm.ac.il; 2Department of Biochemistry, Faculty of Medicine, The Arab American University, Jenin P.O. Box 240, Palestine; bashar.saad@aaup.edu

**Keywords:** obesity, diabetes, hypertension, cardiovascular disease, inflammation

## Abstract

Obesity is a long-term condition resulting from a continuous imbalance between the amount of energy consumed and expended. It is associated with premature mortality and contributes to a large portion of the global chronic disease burden, including diabesity, cardiovascular disease, hypertension, and some cancers. While lifestyle changes and dietary adjustments are the primary ways to manage obesity, they may not always be sufficient for long-term weight loss. In these cases, medication may be necessary. However, the options for drugs are limited due to their potential side effects. As a result, there is a need to identify safe and effective alternative treatments. Recently, dietary compounds, plants, and bioactive phytochemicals have been considered as promising sources for discovering new pharmacological agents to treat obesity and its related complications. These natural products can function independently or synergistically with other plants to augment their effects at various levels of the body. They can modulate appetite, lipase activity, thermogenesis and fat synthesis and degradation, satiation, adipogenesis, and adipocyte apoptosis. Additionally, targeting adipocyte growth and differentiation with diverse medicinal plants/diet is a significant strategy for devising new anti-obesity drugs that can intervene in preadipocytes, maturing preadipocytes, and mature adipocytes. Clinical trials have shown that the wild edible plants in the Mediterranean diet can reduce the risk of obesity and its related diseases. This review examines the effectiveness of the common components of the Mediterranean diet in managing obesity and its associated health issues. We conducted a comprehensive literature review using PubMed, Science Direct, Google Scholar, and Medline Plus to gather data on the therapeutic effects of the Mediterranean diet and phytochemicals in treating obesity and its associated diseases.

## 1. Introduction

Obesity is a serious public health threat, contributing to many chronic diseases worldwide, particularly type 2 diabetes (T2D), cardiovascular disease (CVD), hypertension, and certain cancers [1,2,3]. In recent decades, cultural globalization and urbanization have led to a shift towards a lifestyle with reduced physical activity and that involves consuming more foods high in refined carbohydrates, salt, saturated fats, and proteins, while eating fewer fruits, vegetables, and wild edible plants (WEPs). Obesity causes adipocyte hypertrophy and hyperplasia, leading to molecular and cellular changes that can affect systemic metabolism. This can result in metabolic syndrome and comorbidities such as T2D, CVD, hypertension, and endothelial dysfunction [4,5,6]. Additionally, the increased weight associated with obesity can cause mechanical problems such as osteoarthritis and sleep apnea, affecting quality of life. Numerous studies have demonstrated that obesity elevates the risk of contracting communicable diseases, especially viral infections [7,8] (Figure 1).

In 2016, 39% of adults worldwide (nearly 2 billion) had a BMI of ≥25 kg/m^2^ (overweight) and 12% (671 million) had a BMI of ≥30 kg/m^2^ (obese) [8]. This represents a threefold increase in obesity prevalence since 1975. If current trends persist, by 2025, almost 20% of the world’s population is expected to be obese. The global increase in obesity among children and adolescents is particularly alarming; more than 7% were obese in 2016, compared to less than 1% in 1975 [1,9]. In 2013, there was a rise in obesity rates in Arab countries. Its rates among children and adolescents varied from 3% to 18% for females and from 5% to 14% for males [10,11].

Similarly, the prevalence of T2D and diabesity in many nations has reached epidemic proportions [12,13,14]. Numerous clinical studies have established a connection between excess weight and an increased likelihood of developing T2D and diabesity [15,16,17]. Compared to individuals with normal BMI values, those with a BMI of 25.0–29.9 kg/m^2^ are 3 times more likely to develop T2D, while those with a BMI of ≥30 kg/m^2^ have a 20 times higher likelihood of developing the condition [18]. Abdominal fat accumulation can worsen insulin resistance, making it a significant risk factor for developing diabesity [19,20].

Adipose tissue is responsible for more than just storing fat and balancing energy. It also produces various molecules that can affect the immune system, such as adiponectin, adipokines, cytokines, and chemokines. These molecules help regulate metabolism and inflammation both locally and throughout the body [21,22,23]. Brown and white adipose and tissues are the main types of adipose tissue. The former burns energy to maintain body temperature. On the other hand, white adipose tissue functions as an energy store, offers insulation against the cold, and provides cushioning to protect the body. Additionally, it serves as an endocrine organ [24,25,26]. It usually produces anti-inflammatory mediators; when cells become too large, they can release pro-inflammatory hormones and cytokines. These include, among others, tumor necrosis factor-alpha (TNF-α), interleukin-6 (IL-6), plasminogen activator inhibitor-1 (PAI-1), angiotensinogen, transforming growth factor-beta (TGF-β), adiponectin, resistin, monocyte chemoattractant protein-1 (MCP-1), and leptin (a hormone that increases inflammation and reduces the release of adiponectin, which reduces inflammation). Visceral fat tissue has a higher rate of fat breakdown and macrophage infiltration and releases more IL-6 and MCP-1 than subcutaneous fat tissue. As obesity levels rise, monocytes migrate into adipose tissue and differentiate into macrophages [27,28], which release pro-inflammatory agents that contribute to chronic low-grade inflammation. The latter reduces insulin sensitivity, leading to high blood glucose levels and eventually T2D and related diseases. According to recent studies, fat tissue around the waist releases pro-inflammatory mediators [29,30], which impair insulin sensitivity in obese persons. 

Weight loss has been a key strategy in managing obesity and related health issues. This can be achieved through exercise, diet, and lifestyle changes. However, these methods have not been effective long-term for many overweight and obese individuals, with most regaining their weight within 5 years. Consequently, the search for natural products that can safely and effectively assist in weight loss and the management of obesity-related chronic diseases is on the rise [31,32,33,34]. In traditional medicine, people often turn to herbal and dietary products to help them lose weight. These products contain plant-based compounds that may help prevent obesity and its associated health issues. However, not all natural weight loss remedies are safe or effective. Due to a lack of regulation, some have caused serious health problems, including fatalities [31,32,33,34].

## 2. Prevention and Treatment of Obesity by Mediterranean Dietary Compounds and Wild Plants

In addition to using the whole herb or its extracts, numerous scientific in vitro and clinical trials have confirmed the anti-obesity activities of phytochemicals [33,35,36,37]. The pharmacological mechanisms of the anti-obesity effects of active compounds from pomegranate, citrus fruits, rosemary, black seeds, cumin, ginger, olive leave/oil, turmeric, cinnamon, fenugreek, and garlic have been evaluated through tests on cells, animals, and humans [33,34,35,36,37]. Dietary compounds, medicinal plants, and phytochemicals generated from them were reported to fight obesity by affecting different processes in the body. These include the blocking of pancreatic lipase and glucosidase, the suppression of hunger, the stimulation of thermogenesis and lipid metabolism, and the inhibition of fat breakdown and adipogenesis (Figure 2). Phytochemicals have been found to target fat cells in several ways, including inhibiting fat cell formation, promoting fat breakdown, and reducing energy intake while increasing energy expenditure (Figure 3). These findings come mainly from laboratory and animal studies. However, more high-quality research is needed to confirm the effectiveness of these phytochemicals in humans. Some examples of polyphenols include quercetin, myricetin, diadzein, genistein, cyanidin, luteolin, apigenin, kaempferol, proanthocyanidin, xanthohumol, and epigallocatechin gallate (EGCG). Numerous scientific studies demonstrated that these compounds exhibit weight-reducing and anti-obesity effects [33,35,36,37,38,39] (Figure 2).

## 3. Wild Edible Plants and Their Active Compounds

Due to their rich content in plant secondary metabolites, including polyphenols and terpenoids, WEPs are prime candidates for use in nutraceuticals or functional foods. The Mediterranean area is renowned for its abundant variety of WEPs that are edible and form a vital component of this diet. Local communities have long acknowledged the nutritional, protective, and medicinal benefits of these plants, even before their advantages were scientifically validated. In the eastern Mediterranean, WEPs continue to be valued as healthy food sources and are often collected by women as a means of subsistence and income generation in rural areas with limited economic opportunities [40,41,42]. The scientific community has recently shown an increased interest in traditional Arab-Islamic herbal medicine, particularly in its potential for treating metabolic and chronic diseases [38,39]. This practice, known as Greco-Arab medicine, remains dominant in the Mediterranean region as well as in many Arab and Islamic countries. According to surveys, the Middle East is home to over 2600 species of plants, of which more than 700 are utilized in treating various ailments. Currently, Arab traditional medicine uses fewer than 200–250 plant species to treat various diseases. Many of these species are WEPs that have been assessed in cell culture, in vivo, and in clinical trials and were found to contain pharmacology active compounds [43]. WEPs have similar effects as medicinal plants. They can modulate appetite, lipase activity, thermogenesis and fat synthesis and degradation, satiation, adipogenesis, and adipocyte apoptosis. Additionally, targeting adipocyte growth and differentiation with diverse medicinal plants/diet is a significant strategy for devising new anti-obesity drugs that can intervene in preadipocytes, maturing preadipocytes, and mature adipocytes.

Many plant extracts from the Mediterranean region and their natural compounds have been tested for their anti-obesity effects in the past ten years [33,34,41]. Specifically, 31 plant species showed the ability to block the pancreatic lipase enzyme. The flavonoid rutin (found in citrus and in a wide range of herbs) and the phenolic acids p-coumaric (found in a wide variety of herbs) and ferulic acids (an antioxidant found in a wide range of plants), which are three bioactive compounds, were found to be successful in blocking this enzyme. It was also shown that trans-anethole and resveratrol, two natural compounds, work by stimulating the brown fat tissue [41]. 

In addition, wild cranberries can influence fat cell formation, and blueberries have shown a remarkable ability to lower blood lipids. In addition, 28 plants from the Mediterranean region and 21 active substances from them have shown promising effects on sugar metabolism. Many studies have examined how plant extracts and pure substances can block digestive enzymes related to obesity and T2D (α-amylase, α-glucosidase, and pancreatic lipase). For example, the effects of 18 WEPs that people eat in southern Italy’s Calabria region on pancreatic lipase, an enzyme that breaks down fats, were tested in a recent study [44,45]. Nine of the plant extracts could inhibit the enzyme at a concentration of less than 10 mg/mL. The leaves of purslane *(Portulaca oleracea)* and bladder campion *(Silene vulgaris)*, extracted with water and ethanol, had the strongest inhibition, with 5.48 mg/mL and 6.02 mg/mL needed to block half of the enzyme activity, respectively. Among the plants from the mint family (Lamiaceae), spearmint *(Mentha spicata)* and rosemary (*Rosmarinus officinalis)* showed inhibition, with 7.85 mg/mL and 7.00 mg/mL required, respectively. The most effective plants from the sunflower family (Asteraceae) and the cabbage family (Brassicaceae) were common sowthistle *(Sonchus oleraceus)* (9.75 mg/mL) and perennial wall-rocket *(Diplotaxis tenuifolia*) (7.76 mg/mL), respectively.

Many phytochemicals can help reduce weight by decreasing the absorption of lipids and inhibiting pancreatic lipase. For example, green tea contains catechins and saponins, such as EGCG, which have this effect. Pomegranate (*Punica granatum*) contains punicalagin, ellagic acid, and anthocyanins; rosemary has rosmarinic acid and carnosic acid; black seed contain thymoquinone; and soybean contains proteins and isoflavones that can also help with weight reduction [33,35]. However, most of the studies on how plant extracts affect the body were conducted in vitro or on animals. In addition, many new substances and natural products from plants were found to block pancreatic lipase, an enzyme that breaks down fat, better than orlistat, a common drug for weight loss. Some of these extracts have strong effects on fat digestion because of their compounds like polyphenols, saponins, and terpenes. Many natural products that stop pancreatic lipase are being tested on animals or cells, but none of them have been tested on humans yet. Sometimes, it is hard to apply the findings from these tests to clinical use, because they may not work as well in practice. Therefore, the main problem with these studies is that even though many plant chemicals are stronger than orlistat, we do not know how safe they are compared to orlistat.

When energy intake consistently exceeds energy expenditure, the excess energy is mostly stored as triglycerides in fat tissue. An increase in adipose tissue mass can result from an increase in cell size, cell number, or both. The cellular pathways that control the growth of pre-adipose cells, adipose differentiation, and lipogenesis in adipocytes were thoroughly investigated [1,3,4] (Figure 4). Recently, the differentiation of pre-adipocytes has garnered research attention and has been investigated through in vitro adipogenesis models, including the 3T3-L1 cell line [46,47,48,49]. Many published reports focused on the effects of medicinal plants and their active compounds on adipocyte life cycle. Curcumin (from turmeric), oleuropein (from olive oil), thymoquinone (from black seeds), rosmarinic acid (from rosemary), resveratrol (from grapes), punicalagin (from pomegranate), coumestrol (from soybeans), quercetin (found in many fruits, flowers, and vegetables), Luteolin (found in many fruits, vegetables, and medicinal herbs), and fisetin (found in many fruits and vegetables such as strawberry, apple, and onion) were reported to affect the adipogenesis (Figure 3).

Our bodies maintain an equilibrium of body mass, specifically body lipids, through the coordinated functioning of the autonomic, nervous, endocrine, and metabolic systems. This equilibrium is based on an ‘ideal’ level determined by the central nervous system weight loss. This system can cause 80% of the weight to be regained. Brown adipose tissue plays a key role in energy expenditure by generating heat through thermogenesis. This process is regulated by a metabolic pathway involving the hormone leptin and the protein PGC-1𝛼. This pathway is responsible for burning fat and generating heat, but it can also make it difficult to sustain weight loss due to opposition from the body’s equilibrium. Thermogenesis is a process that mainly takes place in brown adipose tissue but also in beige cells located in white adipose tissue. This process, which is beneficial for preventing obesity, has been shown to be accelerated by capsaicin, caffeine, ephedrine, resveratrol, EGCG, gingerol, and oleuropein. It has been proposed that these compounds could be used to treat obesity and being overweight. For example, caffeine produces thermogenic effects by preventing the degradation of cAMP caused by phosphodiesterase. Caffeine has been shown to decrease food intake and lipid storage. Human studies have also demonstrated that capsaicin, (an active component of chili peppers) and EGCG can increase thermogenesis. Capsaicin treatment stimulates the secretion of catecholamines from the adrenal medulla in a dose-dependent manner, resulting in a thermogenic effect. EGCG increases thermogenesis by inhibiting catechol methyl-transferase, an enzyme responsible for the degradation of norepinephrine [33,41].

## 4. The Mediterranean Diet and Its Active Compounds

The main ingredients of the Mediterranean diet (MedDiet) are among the most researched dietary components for treating and preventing various metabolic disorders and CVD. They are recognized for decreasing the risk of CVD, hypertension, T2D, being overweight/obesity, breast and colon cancers, asthma, and mental decline [33,34,50]. The MedDiet, which is rich in antioxidants and has anti-inflammatory, hypotensive, and hypolipidemic properties, is a great alternative to a diet high in red meat for reducing the risk of CVD. Its positive effects are similar to those of standard drugs such as beta-blockers, aspirin, and angiotensin-converting enzyme inhibitors [50,51,52,53,54,55,56]. However, it is not yet clear whether these benefits come from individual components of the diet or their combined and synergistic effects [50,51,52,53].

There is a lot of evidence that shows a lower risk of mortality from CVD, T2D, specific types of cancer, and cognitive problems when following a MedDiet [53,54]. A comprehensive review of 27 meta-analyses based on 70 cohort studies found 34 different ways to measure the MedDiet [57]. There are various ways to evaluate the MedDiet [58,59], and all of them involve the common food groups that define this dietary pattern. These include a high intake of fruits, vegetables, nuts, legumes, fish, whole grain cereals, and extra virgin olive oil; a moderate consumption of alcohol, preferably red wine; and a low consumption of dairy products, red meat, and processed meat. 

Recent meta-analyses of observational studies have consistently shown that following a MedDiet is associated with positive health outcomes [60,61,62]. Critical reviews of both observational studies and RCTs have also confirmed the health benefits of this diet [63,64,65,66]. A recent Cochrane review concluded that the evidence supporting the effectiveness of the MedDiet in preventing CVD is only of low to moderate certainty. A recent Cochrane review concluded that the evidence supporting the effectiveness of the MedDiet in preventing CVD is only of low to moderate certainty [63]. Nutrition research RCTs are often limited by factors such as small sample sizes, high dropout rates, and short follow-up periods, whereas larger samples, lower dropout rates, and longer follow-up periods are needed to observe patient-relevant outcomes [64]. Some discrepancies may arise from the use of varying definitions of the dietary pattern being studied. Inconsistencies may also result from differences in the diets being compared. 

The health benefits of the MedDiet may be attributed to its impact on the composition and metabolism of the gut microbiota. A systematic review was recently conducted to investigate the effects of this diet on the gut microbiota, as observed in both observational studies and randomized controlled trials. While some research has indicated that a MedDiet may have a positive effect on certain microbiota, a systematic review found that this diet did not consistently alter the composition or metabolism of the microbiota. This inconsistency may be attributed to variations in methodology among studies, particularly in the composition of the MedDiet [65]. 

## 5. The MedDiet and Cardiovascular Disease

A recent long-term RCT (CORDIOPREV study) from Córdoba, Spain, with a 7-year follow-up, compared the effects of the Mediterranean and low-fat diets on the secondary prevention of CVD based on cardiovascular outcomes in patients with coronary heart disease [66]. The study involved 1002 patients, with 500 assigned to a low-fat diet group and 502 to a MedDiet group. The primary outcome was a composite of major cardiovascular events, including myocardial infarction, revascularization, ischaemic stroke, peripheral artery disease, and cardiovascular death. The primary endpoint occurred in 198 participants: 87 in the MedDiet group and 111 in the low-fat group. Multivariable-adjusted hazard ratios of the different models ranged from 0.719 to 0.753 in favor of the MedDiet. These effects were more evident in men, with primary endpoints occurring in 16.2% of the men in the MedDiet group versus 22.8% of the men in the low-fat diet group. These data suggest that, in terms of preventing major cardiovascular events in secondary prevention, the MedDiet was more effective than a low-fat diet [66].

### 5.1. Olive Oil

Olive products are common in the MedDiet and contain polyunsaturated fatty acids, antioxidants, and other nutrients that have beneficial cardiovascular properties. In addition, they contain many healthy active compounds like apigenin, elenolic acid, hydroxytyrosol, ligstroside, oleoside, oleuropein, oleuropein aglycone, and tyrosol. Oleuropein is a polyphenol that is mostly found in olive leaves and young olive fruits. They can improve blood flow, lower inflammation, prevent clots, as well as lower blood pressure and cholesterol and blood glucose levels. A large amount of the available scientific reports emphasize the pharmaceutical potential of oleuropein to treat obesity, diabetes, cardiovascular complications, neurodegenerative diseases, cancer, inflammation, microbial infections, hypertension, and oxidation [67,68,69,70]. 

Olive leaves have proven benefits for lowering cholesterol and blood pressure [70,71,72]. A safe and stable olive leaf extract (EFLA^®^943) was tested on animals and humans for its effects on blood pressure. In a small clinical trial in Germany, 20 pairs of identical twins with mild hypertension took EFLA^®^943 (500 or 1000 mg daily for 8 weeks) or a placebo. The results showed that EFLA^®^943 significantly reduced both systolic and diastolic blood pressure in the twins, more than with lifestyle changes alone. The extract also lowered LDL-cholesterol levels at both doses [73,74]. Another RCT study compared the effects and safety of olive leaf extract and Captopril, a blood pressure drug, in people with mild hypertension [75]. The participants took either 500 mg of olive leaf extract (EFLA^®^943) or 12.5 mg of Captopril twice a day for 8 weeks, after a 4-week run-in period. Both groups showed significant reductions in systolic and diastolic blood pressures after 8 weeks, with no major differences between them. The systolic blood pressure dropped by 11.5 ± 8.5 mmHg in the olive group and by 13.7 ± 7.2 mmHg in the Captopril group, while the diastolic blood pressure dropped by 4.8 ± 5.5 mmHg and 6.4 ± 5.2 mmHg, respectively, from the baseline to the end of the study. The olive group also had a significant decrease in triglyceride levels, which was not seen in the Captopril group [75].

In a Mediterranean population of adults, consuming more olive oil was linked to a reduced risk of death from all causes, CVD, and cancer over the long term. A recent clinical study, conducted over 18 years, evaluated the link between regular olive oil intake and mortality from all causes, cardiovascular disease, and cancer among adults in Spain. Olive oil consumption was assessed in 1567 participants aged 20 years and older from the Valencia Nutrition Study in Spain. Consuming olive oil was linked to a reduced risk of death from all causes, CVD, and cancer. When compared to eating it less than once per month, eating up to one or two tablespoons per day was associated with a 9% and 31% lower risk of all-cause mortality, respectively. Furthermore, the consumption of two or more tablespoons per day was also associated with a lower risk of mortality for CVD and cancer (Figure 5) [76]. Two other studies examined how olive oil intake influenced the risk of developing coronary heart disease and stroke. One study in Spain found that people who ate more than 28.9 g of olive oil a day had a 22% lower risk of coronary heart disease. Another study in France found that people who used olive oil a lot had a 41% lower risk of stroke than those who did not [73,74,77,78,79,80]. Another multicenter trial in Spain was conducted with participants who had no CVD when they enrolled but were at high risk of cardiovascular problems. They were either men aged 55 to 80 or women aged 60 to 80. They had T2D or at least three of these risk factors: smoking, hypertension, low HDL levels, high LDL levels, overweight/obesity, or a family history of coronary heart disease. The participants followed one of these diets: a MedDiet with extra-virgin olive oil or mixed nuts, or a controlled diet with reduced fat intake. The main endpoints were myocardial infarction, stroke, and death from cardiovascular causes. After a median follow-up of 4.8 years, the trial was concluded based on an interim analysis. The findings revealed that a MedDiet with no restrictions, supplemented with olive oil or nuts, significantly decreased the occurrence of major cardiovascular events in adults at high risk. This reinforces the MedDiet’s well-established reputation for reducing the risk of CVD [73,74,77,78,79,80].

Olive oil-derived oleuropein was found to regulate energy metabolism and adiposity [81,82], lowering the levels of PPARγ, compromising adipocyte differentiation, and mitigating insulin sensitivity [83]. Another interesting mechanism studied by Oi-Kano et al. in experimental models showed an increase in uncoupling protein 1 (UCP1) expression, which translates to the formation of “beige” adipose tissue, leading to a decrease of visceral fat mass [84]. Hydroxytyrosol and its derivatives account for over 90% of olive oil’s total polyphenol content [85]. Hydroxytyrosol reduces adipocyte size by downregulating the expression of PPARα and PPARγ [86]. In addition, adipocytes treated with hydroxytyrosol showed an increase in AMPK and lipase [87]. Furthermore, it was not reported that these effects had any influence on body weight or fat levels in humans [87]. Polyphenols have several mechanisms of action (Figure 2). However, more RCTs are needed to confirm if their antioxidant and anti-inflammatory properties can translate to anti-obesity effects by suppressing oxidative stress and inflammation.

### 5.2. Fruits and Vegetables

Eating fruits and vegetables regularly is common advice in most MedDiets. The WHO advises people to eat at least 400 g of fruits and vegetables every day [88]. Several studies showed that eating 800 g of fruits every day could lower the chance of getting CVD by 27% [61,88,89,90,91] and that women who ate more fruits with flavonoids, such as strawberries and grapefruit, had less risk of dying from heart problems [92,93]. Fruits can also help with other health issues that affect the heart. For example, berries with anthocyanins and procyanidins can help with blood vessel problems, blood fat levels, blood clotting, and blood pressure [92,93,94,95]. Orange and other citrus fruits with flavanones can help with high cholesterol [94]. Eating cherries can lower cholesterol and inflammation levels [96]. Drinking juice from fruits like oranges, berries, and cherries can reduce blood pressure and cholesterol [94]. More fruits and vegetables can also help with weight loss, which is another way to improve cardiovascular health [70,73,74,75]. In this regard, several studies in the Mediterranean area have shown that eating more fruits and vegetables is linked to a lower blood pressure and BMI and a reduced risk of death from ischemic heart disease. For instance, a 2004 study of a group of people who were followed over time found that those who ate more fruits and vegetables had a lower blood pressure in a community that consumed a lot of fat from plants. Another study found that increasing fruit and vegetable intake is associated with a lower BMI. Despite some limitations in these studies, a large 2006 analysis of more than 200,000 patients suggested that each serving of vegetables reduced the risk of heart and blood vessel diseases by 4%, and each serving of fruit reduced it by 7%. Another large analysis of studies (almost 200,000 patients) found that eating 3–5 servings of fruits and vegetables per day lowered the risk of heart and blood vessel diseases by 17%. After following 313,074 patients for 8 years who did not have a hardening of the arteries, evidence from a European study on cancer and nutrition showed that those who ate eight servings of fruits and vegetables per day had a 22% lower risk of dying from a lack of blood flow to the heart than those who ate three servings or less [95,96].

### 5.3. Tomatoes and Lycopene

Tomatoes, which are an essential part of the MedDiet, are rich in lycopene, a carotenoid that has been associated with a lower risk of cancer, CVD, cognitive decline, and osteoporosis (Figure 6). By increasing the levels of antioxidants in the body, tomatoes help to reduce oxidative stress and provide protection against various diseases. They also contain phenolic compounds such as quercetin, kaempferol, naringenin, caffeic acid, and lutein [97]. Several epidemiological studies support the traditional benefits of lycopene in preventing CVD. Lycopene’s benefits come from its ability to act as an antioxidant scavenger and inhibit pro-inflammatory and pro-thrombotic mediators. While many aspects of lycopene’s metabolism and functions remain unknown, low doses have been reported to prevent various aspects of CVD [52,98,99]. A systematic review and meta-analysis by Cheng et al. found that increasing tomato and lycopene intake improved blood pressure and endothelial function [100]. A double-blind trial by Gajendragadkar et al. studied the effect of lycopene on CVD patients and healthy volunteers [101]. In this 2-month trial, 36 CVD patients and 36 healthy volunteers were given either 7 mg/day of lycopene or a placebo (NCT01100385). The main result observed was the function of endothelium, which was assessed by analyzing the reactions of the forearm to infusions of acetylcholine into the arteries. Additional outcomes encompassed arterial rigidity, blood pressure, and concentrations of oxidized low-density lipoprotein (ox-LDL), high sensitivity C-reactive protein, and cytokines. In patients with cardiovascular disease, lycopene enhanced endothelial function, but no such improvement was observed in healthy individuals. Blood pressure, arterial stiffness, lipids, and high sensitivity C-reactive protein levels were not affected in either group. The authors concluded that lycopene, a vital component of the MedDiet, improved endothelial function in at-risk individuals [101].

### 5.4. Nuts and Seeds

Eating nuts and seeds may lower the risk of cardiovascular disease, particularly chronic heart disease, possibly by improving blood lipid levels [102]. According to several studies [102,103,104], consuming nuts is linked to a lower risk of T2D, hypertension, and artery disease. Additionally, nut intake has been shown to reduce CVD risk factors such as fatty acids, total cholesterol, and LDL levels [102]. As previously stated, a nut-rich MedDiet has been associated with a decreased risk of CVD by numerous RCTs. Many cohort studies and some smaller RCTs imply that changes in lipid profiles, reactive oxygen species, and blood pressure are to blame for the decrease in CVD morbidity and mortality. Furthermore, the significant evidence for the advantages of daily nut consumption on CVD prevention shows that it may be helpful even in the absence of the additional benefits of the MedDiet. Combining nut consumption with the MedDiet, on the other hand, may give even higher cardiovascular advantages [102,104].

Almonds are known for their cardiovascular benefits, but the exact ways in which they work are not fully understood. They contain monounsaturated fat, fiber, and polyphenols such as tannins and flavonoids. These polyphenols are bioavailable and can be transformed by the body and its microbiota [105,106]. They have been shown to reduce inflammation and oxidative stress. Almonds are also rich in α-tocopherol, a type of vitamin E that acts as an antioxidant and can help protect against obesity, metabolic syndrome, and high lipid levels [107,108,109,110,111]. According to a meta-analysis of studies on people with normal lipid levels, prediabetes, diabetes, obesity, and/or high lipid levels, eating almonds can lower LDL levels without affecting HDL levels [112]. Additionally, an RCT on patients with coronary artery disease found that almonds can raise serum HDL levels [104]. In patients with coronary artery disease, having low levels of HDL cholesterol is a risk factor for cardiovascular disease, even if their LDL cholesterol levels are within the normal range. In fact, nearly half of the hospitalized coronary artery disease patients have normal LDL but low HDL levels. An RCT was conducted to evaluate the impact of almonds on HDL levels in coronary artery disease patients with low initial HDL levels [113]. This study found that eating almonds significantly increased HDL cholesterol levels by 12–16% after 6 and 12 weeks compared to the baseline. Additionally, decreases in total cholesterol, triglycerides, LDL and VLDL cholesterol levels, total-to-HDL and LDL-to-HDL ratios, and atherogenic index were observed after 6 and 12 weeks. These findings indicate that eating 10 g of almonds daily before breakfast can enhance HDL levels and other markers of lipid profiles in patients with coronary artery disease and low HDL [113,114,115].

A recent study by Osorio-Conles et al. [105] evaluated the effects of a MedDiet supplemented with almonds on obesity-related white adipose tissue dysfunction. The study involved 38 women with obesity who were randomly assigned to either follow the diet or continue their usual eating habits for 3 months. The results showed that the almond-supplemented diet increased the abundance of small adipocytes and upregulated the expression of genes related to angiogenesis, adipogenesis, autophagy, and fatty acid usage. The diet also increased the presence of PPAR, endothelial cells, M2-like macrophages, ADRB1, and UCP2 proteins compared to the controls. These changes were associated with a reduction in pro-inflammatory mediators and LDL cholesterol levels, suggesting that including almonds to the MedDiet may ameliorate obesity-related white adipose tissue dysfunction [105]. 

## 6. Anti-Obesity Effects of MedDiet Polyphenols and Their Possible Mechanisms of Action

The polyphenol rich components of the MedDiet are responsible for the health benefits of this diet. In general, polyphenols are classified into two groups: flavonoids and non-flavonoid polyphenols. Members of both groups can exist in their free form or combined with sugars or acylated sugars (glycosides) or amides, esters, and methyl ethers. Flavonoids include around 6000 different chemicals and are classified into several subgroups [116,117]. Hydrolysable tannins, lignans, stilbenes, and phenolic acids belong to non-flavonoid polyphenols that have a more complex structure [118,119,120]. Regular consumption of polyphenols is associated with a lower blood pressure and adiposity, an improved lipid profile, as well as antioxidant and anti-inflammatory effects, all of which help protect against CVD [81,121,122]. Polyphenols may contribute to weight loss through several mechanisms. These include promoting satiety, stimulating thermogenesis by activating brown fat, regulating fat tissue by inhibiting fat cell growth and encouraging fat cell apoptosis, and controlling the β-oxidation [123,124,125] (Figure 2). 

Evidence regarding the impact of polyphenols on obesity and related complications in humans is inconsistent. This is due to variations in study designs, populations, intervention periods, and polyphenol supplements. A systematic review of five RCTs compared the MedDiet with low-fat diets, a low-carbohydrate diet, and the American Diabetes Association (ADA) diet [126]. The results show that the MedDiet was more effective for weight loss than low-fat diets but had similar results to the other two interventions. It is unclear if following a traditional MedDiet leads to a reduction in body weight and waist circumference. A meta-analysis of 16 RCTs, however, found that greater adherence to the MedDiet resulted in more weight reduction when compared to a control diet [126,127]. The MedDiet may promote weight loss due to its high fiber content, low energy density, and low glycemic load. The impact on body weight was more significant when the MedDiet was combined with a calorie-restricted plan or increased physical activity. In T2D patients, the Mediterranean-style diet was found to decrease BMI compared to the control diets [127,128].

While some clinical trials have found that polyphenol-enriched foods can decrease body fat mass, they have not shown reductions in body weight, BMI, or waist circumference [129]. In contrast, a recent study using a polyphenol supplement found significant reductions in these measures. Few studies have investigated the link between total dietary polyphenol intake and weight control. One long-term study found that a higher total polyphenol excretion was associated with a lower BMI, body weight, and waist circumference [130,131,132]. In a 14-year longitudinal study of 4280 participants aged 55–69 in the Netherlands, it was found that women who had a higher intake of flavonoids experienced a lower increase in BMI [133]. 

### 6.1. Resveratrol

Resveratrol, a phenolic molecule present in grapes and some berries, has been shown to have favorable anti-obesity properties. It inhibits adipocyte differentiation and proliferation, induces apoptosis, reduces lipogenesis, and boosts lipolysis and β-oxidation [133,134]. The evidence regarding the impact of resveratrol consumption on weight loss and its maintenance is limited, and the effects appear to be achieved only through dietary supplementation. As a result, the anti-obesity potential and the optimal dosage of resveratrol still need to be investigated. Several RCT studies by Tome-Carneiro et al. [135] investigated the CVD properties of a grape supplement that was rich in resveratrol and other grape polyphenols. The results show statistically significant effects on CVD risk factors such as a reduction in LDL-cholesterol, oxidized LDL, and thrombogenic plasminogen activator inhibitor type 1 (PAI-1), as well as an increase in adiponectin and anti-inflammatory cytokines. However, the effects on adiposity parameters were not significant [134,135]. Resveratrol may work by regulating β-oxidation by increasing the activity of AMPK, which regulates glucose transport and fatty acid metabolism [136]. This leads to an increased fatty acid oxidation and enhanced insulin sensitivity. Resveratrol may potentially influence PPAR expression or enhance oxidation by blocking malonyl-CoA production [137,138].

### 6.2. Curcumin

Curcumin, a yellow polyphenol generated from turmeric, is well-known for its multiple health advantages. Its anti-inflammatory, anti-carcinogenic, anti-obesity, antiangiogenic, and antioxidant effects are among them. Curcumin has anti-obesity properties similar to resveratrol. It works by inhibiting adipogenesis and lipid metabolism, diminishing the synthesis of pro-inflammatory cytokines in adipose tissue, and promoting β-oxidation [139]. Like resveratrol, there are limited clinical trials on the anti-obesity properties of curcumin. However, one study found that ingesting 10 mg of curcumin extract daily for 30 days improved the blood lipid profile. This includes a rise in HDL cholesterol and ApoA while decreasing LDL, ApoB, and the ApoB/ApoA ratio [123,140]. Curcumin is high in polyphenols, and there is substantial evidence to support its potential to increase β-oxidation, decrease fatty acid production, and decrease fat accumulation [123]. 

### 6.3. Nigella sativa

*Nigella sativa* and its active component, thymoquinone, have been reported to have numerous medical applications, including anti-inflammatory, antioxidant, immunomodulatory, anti-cancer, anti-diabetic, anti-obesity, hypotensive, anti-nociceptive, anti-histaminic, and anti-microbial effects [31,32,37]. Black seed and thymoquinone have been shown to ameliorate the effects of obesity-related low-grade inflammation by stimulating the proliferation and function of natural killer cells, monocyte function, T-cell-based immunity, and macrophage activity [31,32,37]. A meta-analysis of 14 clinical trials investigated the effects of long-term *Nigella sativa* supplementation on body weight in individuals with metabolic disorders, such as diabetes, prediabetes, autoimmune hypothyroidism, non-alcoholic fatty liver disease, and obesity. The studies used oral *Nigella sativa* capsules containing either powder or oil. Ten of the 14 studies found significant reductions in body weight, BMI, and waist circumference, with more consistent results observed with *Nigella sativa* oil [141]. A recent meta-analysis of 11 studies examined the anti-obesity effects of *Nigella sativa*. The results show that *Nigella sativa* supplement reduced body weight (−2.11 kg), BMI (−1.16 kg/m^2^), and waist circumference (−3.52 cm) significantly compared to the placebo groups. *Nigella sativa* exerts a mild effect on the reduction of BMI and waist circumference. The weight loss mechanisms of *Nigella sativa* may include appetite suppression, decreased glucose absorption, reduced insulin secretion, and increased adiponectin levels. However, these studies were heterogeneous and of limited quality, so the findings should be interpreted with caution. No serious side effects were reported with *Nigella sativa* supplementation. More research is needed to determine its effects on other obesity-related measures [142]. 

### 6.4. Citrus Fruits

Citrus fruits are an essential component of the Mediterranean diet. They contain bioactive compounds that are beneficial for cardiovascular health. These include alkaloids, flavonoids, tannins, phenols, vitamins C, B1, B2, and B3, glycosides, coumarin glycosides, folic acid, some organic acids, essential oils, and saponins. These compounds are known to have many health benefits. Citrus fruits are a great source of flavonoids, which are compounds that have many health benefits. Some of the flavonoids found in citrus fruits include hesperidin, naringin, diosmin, quercetin, and rutin, among others. These flavonoids can provide a range of benefits when consumed as part of a healthy diet. Flavonoids can be found in a variety of citrus fruits, including bergamots, grapefruits, lemons, limes, mandarins, oranges, and pomelos [143,144].

Several studies have reported on the health benefits of citrus flavonoids. These flavonoids, found in citrus fruits, have a range of biological activities that can positively impact human health. Research suggests that citrus flavonoids may help protect against cardiovascular disease (CVD). These flavonoids, found in citrus fruits, have been shown to reduce oxidative stress, hyperlipidemia, and inflammation, as well as improve endothelial function, arterial blood pressure, and lipid metabolism. These effects may contribute to their therapeutic role in preventing atherosclerosis and CVD [145,146]. 

Citrus flavonoids also help improve glucose tolerance and insulin sensitivity. They have been shown to be effective in improving glucose tolerance, increasing insulin secretion and sensitivity, reducing insulin resistance, decreasing liver glucose output and intestinal glucose absorption, enhancing peripheral glucose uptake, suppressing inflammation, and modulating the activity of enzymes and transporters involved in glucose and lipid metabolism in experimental diabetes models [147]. Furthermore, citrus flavonoids regulate fat metabolism and fat cell development, reduce inflammation, and ameliorate endothelial dysfunction. These properties make them effective in promoting overall health and preventing CVD. The intake of citrus flavonoids has been associated with improved cardiovascular outcomes. While citrus flavonoids have been found to have multiple beneficial effects, the exact ways in which they work are not yet fully understood [148]. 

One example of beneficial cardiovascular health bergamot, a citrus fruit from southern Italy, is known for its traditional uses in improving immune response and cardiovascular function. This fruit contains a variety of phytochemicals, including brutieridin and melitidin, as well as other flavonoids, flavones O-glucosides, and C-glucosides [149]. These compounds are believed to contribute to the beneficial cardiovascular health effects of bergamot. Several clinical trials have shown that taking bergamot orally in different forms can lower total cholesterol and low-density lipoprotein cholesterol levels. In vitro studies have also demonstrated that bergamot’s polyphenols can affect the function of AMPK and pancreatic cholesterol ester hydrolase (pCEH) [148,149,150]. Bergamot has been used in numerous clinical trials and has consistently been shown to be well tolerated in studies lasting from 30 days to 12 weeks. For example, a randomized double-blind placebo-controlled study by Toth et al. [151] assessed the hypolipidemic effects of a bergamot polyphenol fraction. The bergamot polyphenol fraction decreased total cholesterol from 262 to 196, LDL cholesterol from 175 to 116, and triglycerides from 252 to 170. In addition, decreases in serum glucose were observed with the bergamot polyphenol fraction from 120 to 98. No changes were observed in the placebo group [149]. 

### 6.5. Epigenetics Mechanisms

Recent research has revealed that, in addition to the mechanisms previously described, polyphenols can act through epigenetic mechanisms, such as acetylation, methylation, miRNAs, ubiquitylation, phosphorylation, and chromatin packaging [33]. MicroRNAs (miRNAs) are small, non-coding RNAs that play a role in regulating gene expression and are involved in controlling both physiological and pathological processes, such as development and cancer. In addition to other external stimuli, the expression of miRNAs can also be influenced by various nutrients, including vitamins, lipids, and phytochemicals [33]. Polyphenols have been found to modulate over 100 miRNAs, which play a role in regulating various cellular processes such as inflammation and apoptosis. Some of the polyphenols that have been reported to exert their effects through microRNAs include naringenin, apigenin, kaempferol, hesperidin, ellagic acid, and oleuropein [150,151]. The majority of the research on this subject has been conducted in vitro using various cell lines, particularly cancer cell lines. Only a few studies have been carried out on animals. Based on the available data, miRNAs appear to be promising mediators in regulating the biological effects of polyphenols. However, further research is needed to validate miRNA targets, especially under physiologically relevant conditions that take into account the bioavailability of dietary polyphenols.

### 6.6. Daily Consumption of Polyphenols

Food dietary studies have shown that there is a large variability in the consumption of polyphenols. For example, the estimated consumption of flavonoids is about 190 mg/day in the United States, 313 mg/day in Spain, and 454 mg/day in Australia [152,153]. A recent study looked back at the relationship between the average daily intake of polyphenols, adherence to the Mediterranean diet, and body measurements in a group of 250 healthy Greek volunteers aged 18 to 65. The average daily intake of polyphenols was found to be 1905 mg, with the majority of participants having moderate or high average consumption in the past year (67.5% of the sample consumed more than 1000 mg/day). A moderate adherence to the MedDiet, as indicated by a higher MedDiet score, was found to be linked to an increase in the average daily intake of polyphenols. An increase in polyphenol intake and a higher MedDiet score were both associated with a decrease in waist-to-hip circumference. Certain functional foods rich in polyphenols, such as sour cherries and tomatoes, were found to be associated with improved body composition measurements. However, larger epidemiological studies are needed to draw more definitive conclusions about the relationship between polyphenol intake and metabolic disease biomarkers in the general population [154].

## 7. Anti-Obesity Effects of Carotenoids and Their Possible Mechanisms of Action

Many studies have found a link between obesity and low levels of carotenoids in the blood [155,156]. Carotenoids are a type of hydrophobic pigment present in vegetables and fruits, which cannot be synthesized by the human body. Consuming them has been linked to numerous health benefits for humans, including a reduction in overall mortality [157]. One of their key characteristics is their ability to affect oxidative stress and inflammation by interacting with transcription factors [158]. For instance, they can serve as precursors for bioactive derivatives that activate signaling through nuclear hormone receptors. These derivatives, such as retinoids or vitamin A derived from β-carotene, can activate retinoic acid receptors (RARs), which are a type of nuclear hormone receptor [159]. Recent research has led to the discovery of various new metabolic pathways. These pathways are mediated through specific nuclear hormone receptor activation pathways, which were predicted and subsequently confirmed.

There is a strong negative correlation between body mass index (BMI) and the levels of all measured carotenoids in the blood. Additionally, many disorders associated with obesity, such as chronic low-grade inflammation and insulin resistance, also show a strong negative correlation with the levels of carotenoids in the blood [160,161,162]. Consuming a diet high in fat can alter the functions of white adipose tissue, which can affect the way AMPK regulates the breakdown of fats and lipid metabolism in fat cells. By activating AMPK, it may be possible to reduce oxidative stress and inflammation. Consuming carotenoids, either through diet or supplements, has been shown to help reduce complications caused by a high-fat diet [163]. Different types of carotenoids can stimulate the AMPK signaling pathway, activating enzymes, increasing the activity of transcription factors, promoting the conversion of white adipose tissue to brown, and inhibiting the formation of new fat cells. Carotenoids may also improve the levels of certain “homeostatic” factors, such as adiponectin, which may play a role in activating AMPK. Based on these findings, it is recommended that clinical trials be conducted to confirm the effects of carotenoids on the AMPK pathway in long-term treatments, particularly in cases of obesity [163].

Several studies have been conducted to investigate the potential use of carotenoids in managing obesity. However, many of these studies used a combination of carotenoids and vitamins in natural sources, such as fruit juices or plant extracts, making it difficult to determine the specific effects of carotenoids alone [164,165,166]. According to our knowledge, only two clinical trials have been conducted that were randomized, double-blind, placebo-controlled, and investigated the effect of pure carotenoid or xanthophyll supplementation. Canas et al. [167] found that children who were given a mixture of carotenoids (including β-carotene, α-carotene, lutein, zeaxanthin, lycopene, astaxanthin, and γ-tocopherol) for 6 months experienced a decrease in their BMI *z*-score, waist-to-height ratio, and subcutaneous adipose tissue. These positive effects were closely linked to an increase in the concentration of β-carotene in the plasma of children [166]. Another study used a combination of paprika xanthophylls and carotenoids, given to healthy overweight volunteers for 12 weeks. This supplementation resulted in a reduction in the visceral fat area, subcutaneous fat area, and total fat area, as well as the BMI in the group that received the treatment compared to a placebo group [166,167,168].

## 8. Concluding Remarks

Obesity is linked to many chronic diseases. Traditional weight loss methods have not been effective, leading to a search for safe natural products for weight loss. Plants and phytochemicals are considered as potential sources for new obesity treatments. The MedDiet, known for its abundance of plant-based foods, is considered a healthy eating pattern that enhances overall health and quality of life. This diet includes polyphenol-rich foods such as olive oil, walnuts, red wine, vegetables, fruits, legumes, WEPs, and whole grains, which have been shown to have anti-obesity effects. Plant- and diet-derived phytochemicals, polyphenols in particular, can work alone or synergistically to reduce appetite, fat metabolism, and adipocyte growth (Figure 2, Figure 3 and Figure 4). Based on existing evidence, the MedDiet and several WEPs are a recommended dietary approach in the prevention of obesity and its related CVD, T2D, and certain types of cancer (Figure 2). The current level of knowledge assigns potential benefits of WEP/MedDiet constituents in terms of hypolipidemic, insulin-sensitizing, anti-inflammatory, anti-oxidative, and antithrombotic activities. The MedDiet has been shown to have numerous health benefits, including reducing LDL and its oxidation, decreasing the production of pro-inflammatory cytokines, increasing insulin sensitivity by modifying cell membranes, and improving endothelial function by increasing the bioavailability of vasodilatory agents (Figure 5). Fiber, phytosterols, polyphenols such as resveratrol, vitamins, and minerals are also beneficial. The present findings, however, are difficult to interpret because the majority of the data originate from cell culture research and animal tests. Despite the encouraging results, the mechanism of action of plants for obesity and CVD has not been determined, and additional studies are required.

## Figures and Tables

**Figure 1 ijms-24-12641-f001:**
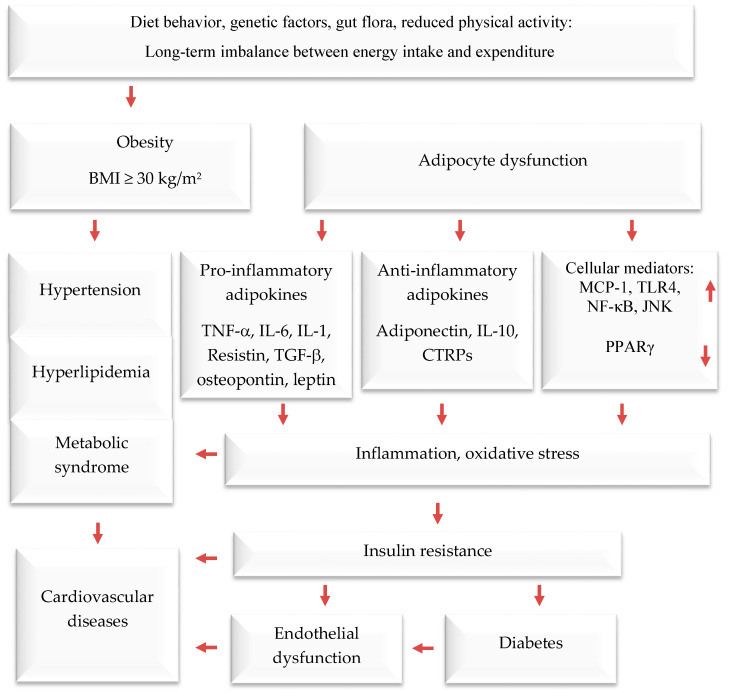
The interconnection between obesity and inflammation, type 2 diabetes, and cardiovascular diseases. Upwards arrows: Activation; downwards arrows: Inhibition.

**Figure 2 ijms-24-12641-f002:**
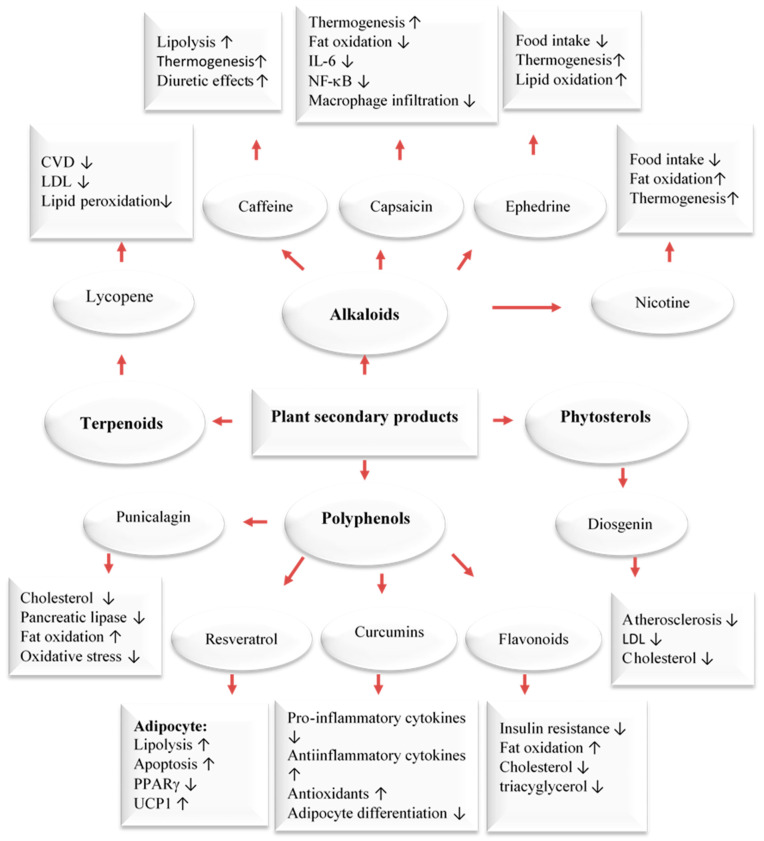
Major classes of plant secondary metabolites and their anti-obesity mechanisms [39]. Upwards: Activation; downwards arrows: Inhibition.

**Figure 3 ijms-24-12641-f003:**
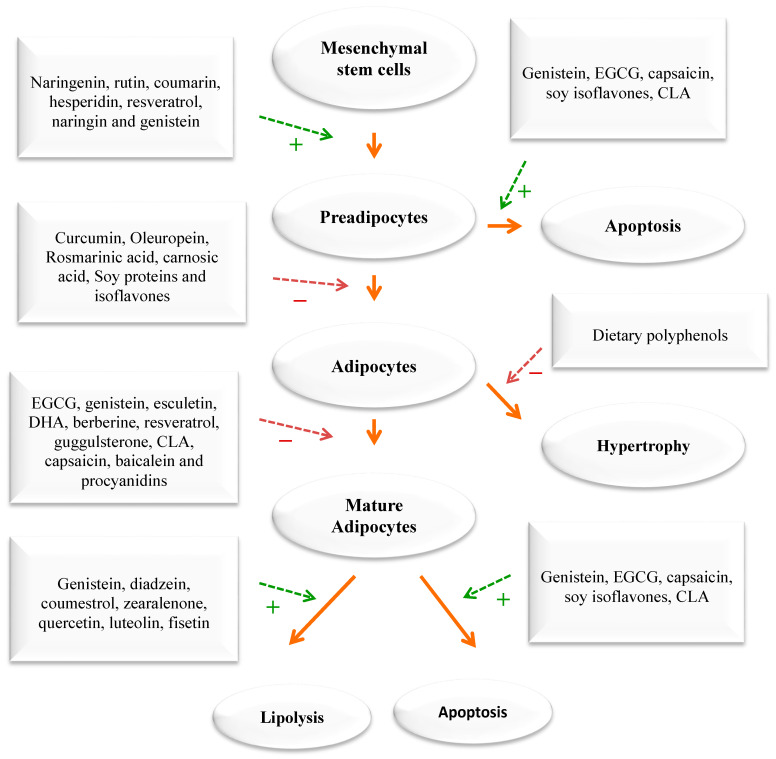
The impact of phytochemicals on adipogenesis. Green dashed arrows: Activation; red dashed arrows: Inhibition.

**Figure 4 ijms-24-12641-f004:**
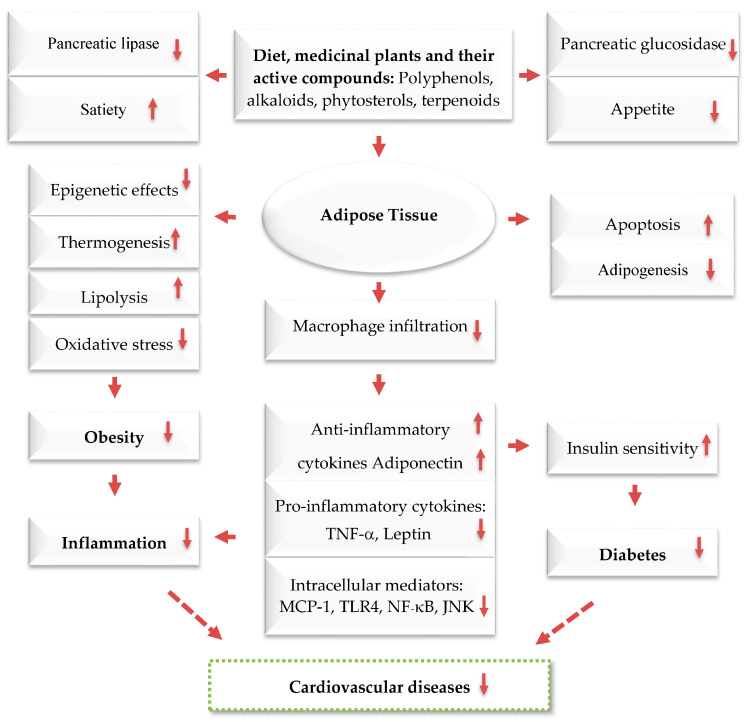
Anti-obesity and its related disease mechanisms of plants and their active compounds. Dashed arrows: Indirect effect; solid arrows: Direct effect; Upwards: Activation; downwards arrows: Inhibition.

**Figure 5 ijms-24-12641-f005:**
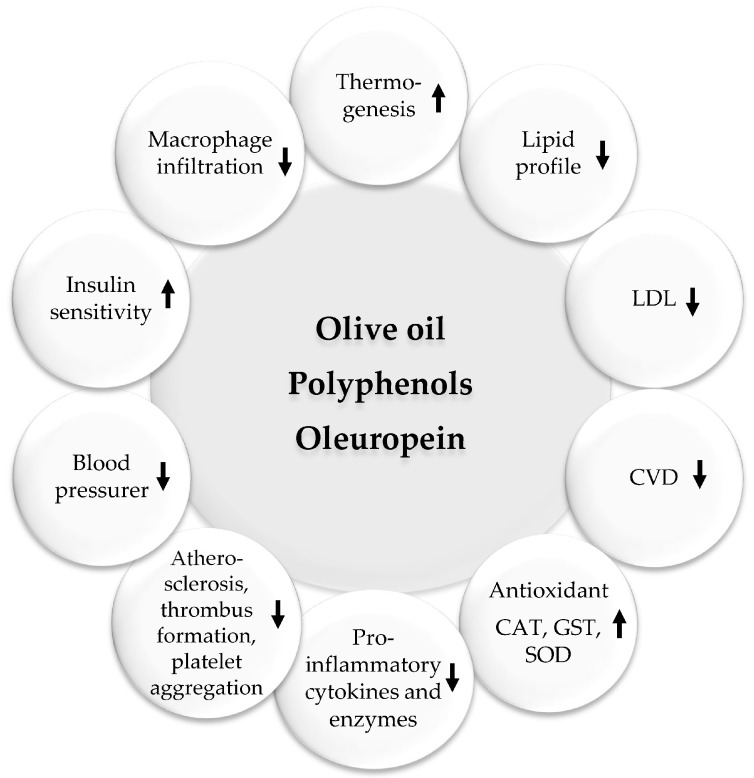
Beneficial pharmacological effects of olive oil. Upwards arrows: Activation; downwards arrows: Inhibition.

**Figure 6 ijms-24-12641-f006:**
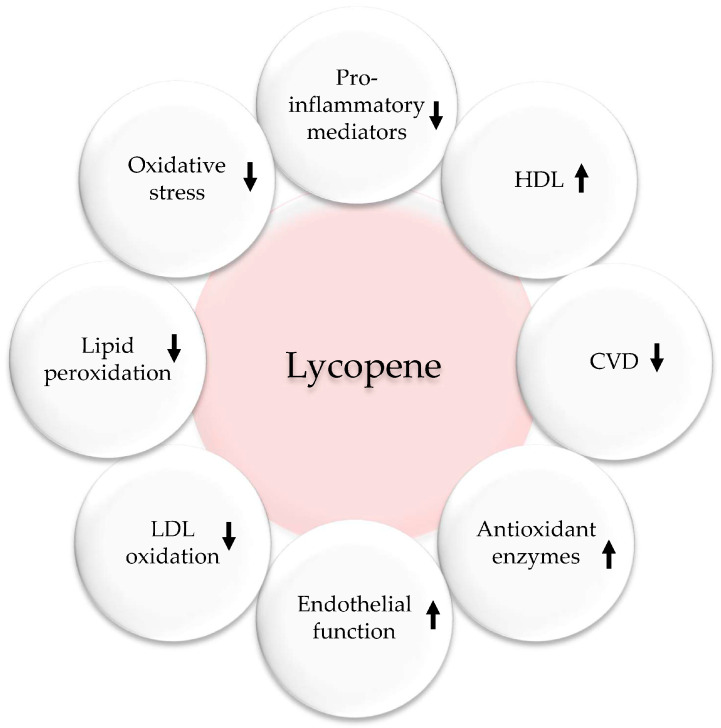
Antioxidant, anti-inflammatory, and cardiovascular effects of lycopene. Upwards arrows: Activation; downwards arrows: Inhibition.

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
