# Peer review of "A Review of the Anti-Obesity Effects of Wild Edible Plants in the Mediterranean Diet and Their Active Compounds: From Traditional Uses to Action Mechanisms and Therapeutic Targets"

_ijms, 2023, doi:10.3390/ijms241612641_

Round 1

Reviewer 1 Report

A review of the anti-obesity effects of wild edible plants in the Mediterranean diet and their active compounds: From traditional uses to action mechanisms and therapeutic targets

This interesting review discussed the effectiveness of the common components of the Mediterranean diet in managing obesity and its associated health issues such as cardiovascular disease (CVD). This review discussed the Wild edible plants, the Mediterranean diet and their active compounds followed by exploring the effects of several foods of the Mediterranean diet on CVD and the relevant mechanism.

The overall impression when reading this review is positive. The chosen topic is interesting and the research design is appropriate. However, the quality of the English language and style needs to be carefully edited. The quality of figures is required to be tremendously reconstructed. Additionally, there are some areas where the text could be further improved.

Abstract:

Line 13: However, the options for drugs are limited due to their low effectiveness and potential side effects.

The description should be carefully modified because not all drugs are of low effectiveness for the treatment of obesity.

1. Introduction

This section needs to be better organized. Most of the content in this section focuses on the prevalence, pathogenesis, and comorbidity of obesity. A more relevant description for this section would review the current evidence on the association between the Mediterranean diet and the risk of obesity and related diseases such as diabetes, CVD, and cancer.

Lines 70-72: It usually produces anti-inflammatory mediators, when cells become too large, they can release hormones and cytokines. These include, among others, tumor necrosis factor-alpha (TNF-α), interleukin-6 (IL-6)…

It is obvious that TNF-a is a typical pro-inflammatory cytokine.

3. Wild edible plants and their active compounds

4. Mediterranean diet and its active compounds

The reviewer’s main concerns are the clarity of the definition of the Mediterranean diet and the molecular mechanism by which the compounds influence obesity and its comorbidities.

Additionally, there are many confusing descriptions, for example:

Line 221-223:  These pharmacological effects might be due to the antioxidant, anti-inflammatory, anti-clotting, anti-cancer, and lipid-lowering effects of the active components of a MedDiet.

Is anti-cancer a molecular mechanism for improving obesity and relevant diseases? Actually, cancer is also an obesity-related disease.

5. MedDiet and cardiovascular disease

Why is CVD discussed only without discussing other serious diseases? For example, some foods are also beneficial for cancer and diabetes. In addition, some foods or vegetables like Olive oil, Fruits and vegetables are common worldwide. It is not unique to the Mediterranean diet.

Why is CVD the only serious disease discussed? Some foods may also have benefits for cancer and diabetes prevention. Moreover, olive oil, fruits and vegetables are not exclusive to the Mediterranean diet. They are widely available around the world.

6. Anti-obesity effects of MedDiet polyphenols and their possible mechanisms of action.

The underlying mechanism of how the Mediterranean diet affects obesity has not been clearly discussed.

The quality of the English language and style needs to be carefully edited.

Author Response

Thank you for your valuable comments. I have carefully considered and addressed all comments and suggestions in my revision. I am open to further revisions if you have any additional requests or suggestions. Below is my point-by-point response to the reviewers’ comments:

Line 13: However, the options for drugs are limited due to their low effectiveness and potential side effects.

The description should be carefully modified because not all drugs are of low effectiveness for the treatment of obesity.

Corrected

  1. Introduction

This section needs to be better organized. Most of the content in this section focuses on the prevalence, pathogenesis, and comorbidity of obesity. A more relevant description for this section would review the current evidence on the association between the Mediterranean diet and the risk of obesity and related diseases such as diabetes, CVD, and cancer.

 Lines 70-72: It usually produces anti-inflammatory mediators, when cells become too large, they can release hormones and cytokines. These include, among others, tumor necrosis factor-alpha (TNF-α), interleukin-6 (IL-6)…

It is obvious that TNF-a is a typical pro-inflammatory cytokine.

Corrected

  1. Wild edible plants and their active compounds
  2. Mediterranean diet and its active compounds

 The reviewer’s main concerns are the clarity of the definition of the Mediterranean diet and the molecular mechanism by which the compounds influence obesity and its comorbidities.

Yes, this correct

 Additionally, there are many confusing descriptions, for example:

Line 221-223:  These pharmacological effects might be due to the antioxidant, anti-inflammatory, anti-clotting, anti-cancer, and lipid-lowering effects of the active components of a MedDiet.

corrected

Is anti-cancer a molecular mechanism for improving obesity and relevant diseases? Actually, cancer is also an obesity-related disease.

Corrected

  1. MedDiet and cardiovascular disease

Why is CVD discussed only without discussing other serious diseases? For example, some foods are also beneficial for cancer and diabetes. In addition, some foods or vegetables like Olive oil, Fruits and vegetables are common worldwide. It is not unique to the Mediterranean diet.

Why is CVD the only serious disease discussed? Some foods may also have benefits for cancer and diabetes prevention. Moreover, olive oil, fruits and vegetables are not exclusive to the Mediterranean diet. They are widely available around the world.

Thank you for your suggestion. While plant products have been shown to have beneficial effects on the management of various diseases, our current review, “A review of the anti-obesity effects of wild edible plants in the Mediterranean diet and their active compounds: From traditional uses to action mechanisms and therapeutic targets,” and the special issue “Natural Products in Obesity: Novel Strategies and Molecular Mechanisms” focus specifically on obesity and cardiovascular disease. As such, our discussion is limited to these topics.6. Anti-obesity effects of MedDiet polyphenols and their possible mechanisms of action.

The underlying mechanism of how the Mediterranean diet affects obesity has not been clearly discussed.

  A new paragraph added: Lines 524-567

Comments on the Quality of English Language

The quality of the English language and style needs to be carefully edited.

Done

Reviewer 2 Report

The manuscript is interesting, even if it takes up a topic already covered, and the attractive graphic schemes that make the review very readable.

There are some points to improve:

- A group of vegetables rich in polyphenols and typical of the Mediterranean maquis has been overlooked, i.e., citrus fruits, oranges, lemons, grapefruit, citrons, and last but not least, bergamot are absolutely to be considered in a work that analyzes the Mediterranean diet.

- In the analysis, miRNAs could be considered influenced by the Mediterranean diet and by fruit and vegetable components (10.3390/antiox10020328, 10.3390/molecules25010063)

- Other compounds could also be considered, such as carotenoids, present in various vegetables typical of Mediterranean cultures (e.g., 10.2174/1389557515666150709112822)

- A suggestion of polyphenol intake during the day would be interesting.

- An emphasis on bioavailability and, when not assimilated, the positive action on the microbiota would be interesting

It needs some revision.

Author Response

Thank you for your valuable comments. I have carefully considered and addressed all comments and suggestions in my revision. I am open to further revisions if you have any additional requests or suggestions. Below is my point-by-point response to the reviewers’ comments:

Comments and Suggestions for Authors

The manuscript is interesting, even if it takes up a topic already covered, and the attractive graphic schemes that make the review very readable.

There are some points to improve:

- A group of vegetables rich in polyphenols and typical of the Mediterranean maquis has been overlooked, i.e., citrus fruits, oranges, lemons, grapefruit, citrons, and last but not least, bergamot are absolutely to be considered in a work that analyzes the Mediterranean diet.

A new paragraph added: Lines 530-572

- In the analysis, miRNAs could be considered influenced by the Mediterranean diet and by fruit and vegetable components (10.3390/antiox10020328, 10.3390/molecules25010063)

A new paragraph added: Lines 576-593

- Other compounds could also be considered, such as carotenoids, present in various vegetables typical of Mediterranean cultures (e.g., 10.2174/1389557515666150709112822)

  A new section added: Lines 613-652

- A suggestion of polyphenol intake during the day would be interesting.

A new paragraph added: Lines 595-611

- An emphasis on bioavailability and, when not assimilated, the positive action on the microbiota would be interesting

Thank you for your suggestion. However, this is a large topic that would be better addressed in a separate review article. The current review is already 35 pages long.

Comments on the Quality of English Language. It needs some revision.

Done

Round 2

Reviewer 1 Report

My major concerns have been addressed.

Reviewer 2 Report

I think that the authors improved the manuscript enough.

Just some other revisions needed.